# Targeted Therapy in Mesotheliomas: Uphill All the Way

**DOI:** 10.3390/cancers16111971

**Published:** 2024-05-22

**Authors:** Elisa Bertoli, Elisa De Carlo, Martina Bortolot, Brigida Stanzione, Alessandro Del Conte, Michele Spina, Alessandra Bearz

**Affiliations:** 1Department of Medical Oncology, Centro di Riferimento Oncologico di Aviano (CRO), IRCCS, 33081 Aviano, Italy; elisa.bertoli@cro.it (E.B.); elisa.decarlo@cro.it (E.D.C.); martina.bortolot@cro.it (M.B.); brigida.stanzione@cro.it (B.S.); alessandro.delconte@cro.it (A.D.C.); mspina@cro.it (M.S.); 2Department of Medicine, University of Udine, 33100 Udine, Italy

**Keywords:** mesothelioma, targeted therapy, molecular alterations

## Abstract

**Simple Summary:**

The search for precision medicine applications in mesotheliomas (MM) is taking its first steps. After platinum and pemetrexed chemotherapy, the treatment for relapsed MM remains an unmet clinical need, and the prognosis of MM remains poor even if the recent introduction of immunotherapy. It is known that MM is mainly characterized by inactivanting tumor suppressor alterations and that these, along with some cellular targets or metabolic enzymes, could be potentially amenable to specific therapies The purpose of this review is to take a comprehensive excursus of the main targets and the related evidence regarding possible treatment activities intended for them.

**Abstract:**

Mesothelioma (MM) is an aggressive and lethal disease with few therapeutic opportunities. Platinum-pemetrexed chemotherapy is the backbone of first-line treatment for MM. The introduction of immunotherapy (IO) has been the only novelty of the last decades, allowing an increase in survival compared to standard chemotherapy (CT). However, IO is not approved for epithelioid histology in many countries. Therefore, therapy for relapsed MM remains an unmet clinical need, and the prognosis of MM remains poor, with an average survival of only 18 months. Increasing evidence reveals MM complexity and heterogeneity, of which histological classification fails to explain. Thus, scientific focus on possibly new molecular markers or cellular targets is increasing, together with the search for target therapies directed towards them. The molecular landscape of MM is characterized by inactivating tumor suppressor alterations, the most common of which is found in CDKN2A, BAP1, MTAP, and NF2. In addition, cellular targets such as mesothelin or metabolic enzymes such as ASS1 could be potentially amenable to specific therapies. This review examines the major targets and relative attempts of therapeutic approaches to provide an overview of the potential prospects for treating this rare neoplasm.

## 1. Introduction

Mesothelial tumors are classified into benign, preinvasive, or mesotheliomas (MMs). MMs are rare tumors affecting mesothelial surfaces, usually the pleura and peritoneum, with pleura as the most common site of MM origin (73–85%) [1]. In this review, we will use the MM abbreviation for mesothelioma (pleural, pericardial, or peritoneal) and MPM for pleural mesothelioma only. Being linked in approximately 80% of cases to exposure to asbestos fibers (chrysotile in 99% of the cases), its incidence is expected to increase worldwide, driven by usage of asbestos in developing countries and long latency periods (damage caused by asbetos through DNA modification and chronic inflammation can take up to 50 years to manifest) [2]. Fluoro-edenite, silicon carbide fibers, carbon nanotubes (MWCNT-7), a prior exposition to high-dose ionizing radiations, and simian virus 40 (SV40) are considererd other possible risk factors, although with lower incidence (approximately 4%) [3]. The WHO classification of MPM recognized three histopathological variants: epithelioid (60%), sarcomatoid (10%), the most aggressive, and biphasic (30%), i.e., the co-presence of both epithelioid and sarcomatoid components [1]. This histopathological classification is prognostically relevant and plays a significant role in treatment decisions [1]. MM is considered a malignancy of the elderly, with a median age at diagnosis of 76 years [4]. It is characterized by a poor prognosis (5-year overall survival rate of 5–10%), presenting in an advanced stage in most cases, and by limited therapeutic armamentarium, with a survival expectancy of 1.5 years from diagnosis [5]. For this reason, in recent years, thanks to the better knowledge of tumor biology and the different underlying molecular mechanisms, with consequent development of targeted therapies in several tumors, there has been a focus on a comprehensive genomic analysis of MM to identify molecular alterations that are potentially susceptible to targeted treatment.

This review aims to summarize and discuss the main data currently available about the potential role of target therapies in the treatment of MM, focusing on the most promising targets and their pathways (Figure 1).

## 2. Current Consolidated Therapeutic Options

The vast majority of patients affected by MPM are treated with systemic treatment due to the lack of consensus about cytoreductive surgery. For decades, the standard of care has been cisplatin and antifolate chemotherapy (CT).

Indeed, in the phase III EMPHACIS trial, overall survival (OS), time to progression (TTP), and response rates (RRs) were superior with pemetrexed plus cisplatin and vitamin supplementation compared with cisplatin alone. Notably, OS improved from 9.3 to 12.1 months, with an increase in RR (41.3% vs. 16.7%) and TTP (5.7 vs. 3.9 months) [6].

Cisplatin can be replaced by carboplatin in elderly or cisplatin-unfit patients with comorbidities [7,8]. A randomized phase III study of cisplatin, with or without raltitrexed, demonstrated superiority for raltitrexed–cisplatin in terms of RR (13.6% vs. 23.6%) and the median OS (11.4 vs. 8.8 months) [9]. Moreover, the combination of standard platinum-based CT and the anti-VEGF bevacizumab was evaluated in the phase III MAPS trial. Despite the improvement, although modest, in OS and progression-free survival (PFS), bevacizumab has never been approved by the FDA and EMA [10]. Platinum-pemetrexed-based CT should be considered at the time of diagnosis before functional clinical worsening and continued for up to six cycles [11]. Maintenance therapy has not demonstrated its efficacy [12].

The introduction of first-line immunotherapy (IO) in 2020 has been the greatest advance in the MM treatment scenario. CheckMate 743, an open-label, randomized, phase III trial, first demonstrated an advantage in the OS of the nivolumab and ipilimumab combination over standard CTs (mOS 18.1 vs. 14.1 months) [13]. The 3-year survival rates were 23.2% in the nivolumab plus ipilimumab arm and 15.4% in the CT arm [14]. The impact of IO appears to be higher in non-epithelioid MM (mOS 18.1 vs. 8.8 months, respectively, in the nivolumab–ipilimumab arm and the CT arm) than in epithelioid histotypes (mOS 18.7 vs. 16.2 months, respectively), probably due to the worst prognosis and lower chemosensitivity of the non-epithelioid histotype [13].

To improve the prognosis of patients with MM even more, several studies developed to evaluate the combination of CT and IO in the first-line setting. Among these, the phase II DREAM study evaluated the addition of durvalumab to cisplatin and pemetrexed for six cycles followed by maintenance with durvalumab [15]. Favorable results, with a 6-month PFS rate of 57%, mPFS of 7 months, and OS of 18.4 months, led to the design of the ongoing phase III DREAM3R study. Another phase II–III trial, the recently published IND.227, showed an improvement of OS for patients treated with the combination of pembrolizumab and CT (mOS 17.3 vs. 16.1 months; 3-year survival 25% vs. 17%) [16].

Unfortunately, most patients’ progress to first-line therapy and the optimal second-line therapy remains an unmet clinical need. The most common option is a platinum-pemetrexed rechallenge if a greater than three months PFS has been obtained after first-line therapy or vinorelbine [17,18]. Due to the recent introduction of nivolumab–ipilimumab, currently available studies regarding second-line treatment involve patients pretreated with CT. Second-line IO has shown conflicting results. In the phase IIb DETERMINE study, Tremelimumab failed to demonstrate a benefit in OS, while pembrolizumab did not confirm the PFS and OS advantage in the phase III PROMISE trial [19,20].

Regarding second- and third-line therapy, in the phase III CONFIRM trial, nivolumab showed an advantage over the placebo, with an mPFS of 3.0 vs. 1.8 months and an mOS of 9.2 vs. 6.6 months [21], and the phase II NIBIT-MESO-1 trial exhibited an RR of 28%, a PFS of 5.7 months, and an mOS of 16.5 months for the durvalumab and tramelimumab combination [22,23]. Moreover, the association between nivolumab and ipilimumab has been explored in the INITIATE study and MAPS-2 study with a response rate of about 35%, a PFS of about 6 months, and an OS over 15 months in both studies [24,25].

In recent years, on the one hand, the introduction of IO in a first-line setting promoted the identification of predictive biomarkers; on the other hand, the absence of standard second-line therapy after CT stimulated the search for new possible molecular targets.

## 3. Potential Molecular Targets for MMs

### 3.1. Gene Involved in Cell Cycle Regulation

#### 3.1.1. CDKN2A and MTAP

The homozygous deletion of 9p21 is a frequent somatic alteration in MPM [26], and it is detected in about 50% to 75% of cases [27]. Because of the proximity relative to chromosome 9p21, the genes involved in this deletion comprise Cyclin-Dependent Kinase Inhibitor 2A (CDKN2A) and methylthioadenosine phosphorylase (MTAP).

##### CDKN2A

CDKN2A encodes for crucial cell cycle proteins: tumor suppressor p16ink4a (a CDK4 and CDK6 inhibitor) and p14ARF (an inhibitor of MDM2 that prevents p53 degradation); moreover, it is the most frequently altered gene in MPM (42–49%) [28,29].

CDKN2A loss and CDK4/CDK6 overexpression are associated with poor prognosis in MPM [30].

Preclinical studies revealed promising results in restoring p16ink4a function with CDK4/6 inhibitors. Abemaciclib and palbociclib showed a significant decrease in cell number (mean of 54.5% ± 5.5 with abemaciclib and mean of 53.4% ± 4.9 with palbociclib), inducing G1 cell cycle arrest and increasing cell senescence. The interferon signaling pathway was also enhanced as a result, favoring the tumor antigen presentation process [31]. The safety and feasibility of this therapeutic approach were evaluated in the single-arm, phase II MiST2 trial. Twice daily, 200mg of abemaciclib was administered to a small group of patients (n = 26) with MM progressed to platinum-based CT. The duration of treatment was 24 weeks, and all patients had p16ink4A-negative disease. Disease control rates (DCRs) were 54% at 12 weeks and 23% at 24 weeks; the median PFS and OS were 128 days and 217 days, respectively. Eight patients exhibited grade ≥ three treatment-related adverse events [32]. In a post hoc exploratory analysis, greater tumor regression was evidenced in patients with concurrent MTAP loss and p16ink4a loss (44% of patients) [32].

These results support CDK4/6 inhibitors as a new possible target treatment for a subgroup of patients with MPM; however, the small sample size of the study and the absence of other phase II or III trials confirming the safety and efficacy of this approach limit the use of CDK4/6 inhibitors in daily clinical practice.

The p14ARF protein promotes the degradation of the MDM2 protein and thus prevents the MDM2-mediated inhibition of p53. Preclinical data suggested the potential role of targeting this pathway for the treatment of MPM. A gene therapy-based approach consisting of adenoviral-mediated p14ARF gene transfection induces apoptotic cell death in human MM cells, due to G1 phase arrest caused by the overexpression of p14ARF [33].

In another phase I clinical study, the oral selective MDM2 inhibitor AMG 232 was investigated, exhibiting acceptable safety and stable disease in solid tumors [34]. AMG232 blocks MDM2–p53 interactions restoring p53 tumor suppression.

##### MTAP

Due to the proximity of the genes on chromosome 9p21, approximately 90% of MPMs with CDKN2A loss also harbor the deletion of MTAP [29]. Immunohistochemical staining for MTAP has become a reliable diagnostic tool in both cytological and histological specimens [35]; MTAP deletion is also considered an adverse prognostic factor, associated with shorter survival [36]. MTAP loss is more common in sarcomatoid MM, in which it is detected in approximately half of the cases [29].

MTAP is engaged in polyamine metabolism. In particular, it encodes a key enzyme of the methionine and adenosine salvage pathway: it catalyzes the phosphorolysis of 5′-deoxy-5′-methylthioadenosine (MTA), an endogenous moderately potent and selective inhibitor of the enzymatic activity of protein arginine methyltransferase 5 (PRMT5) [37]. PRMT5 targets proteins involved in different cellular functions including RNA splicing, transcription, and translation [38]. MTAP loss, with consequent MTA accumulation, renders cancer cells more vulnerable to the further inhibition of PRMT5 [39,40] and can predict sensitivity to target therapies that inhibit PRMT5 or MAT2A [41]. In a phase II study, L-alanosine was tested in MTAP-deficient solid tumors with poor results (no objective responses were observed; only two patients with MPM had prolonged stable disease) [42]. Recently, MRTX1719, a selective inhibitor of PRMT5 in the presence of MTA, demonstrated marked antitumor activity across a panel of xenograft models [43]. The safety and efficacy of MRTX1719 and other MTA-cooperative PRMT5 inhibitors, such as TNG908, are under evaluation.

### 3.2. Gene Coding for Receptor Tyrosine Kinases

#### 3.2.1. EGFR

The epidermal growth factor receptor (EGFR), a member of the ErB family involved in cell growth, proliferation, and angiogenesis, is frequently highly overexpressed in the majority of MPMs, with a reported expression between 44 and 97% [44,45]. Its prognostic role remains controversial.

Until today, numerous EGFR-TKI and monoclonal antibodies have been evaluated in the past years but failed to demonstrate significant clinical efficacy in MPM [46,47,48,49]. The reasons for this are different: despite the overexpression, EGFR kinase domain mutations or EGFR gene amplification are extremely rare in MPM. In addition, the concurrent activation of alternative pathways (e.g., amplification of mesenchymal epithelial transition factor (MET) oncogene and AKT) could stimulate receptor tyrosine kinases, representing a resistance mechanism to EGFR target therapies [50].

New approaches including antibodies targeting EGFRs that deliver cytotoxic CT or miRNA revealed initial promising results in the clinic. In a phase I trial, TargomiRs, minicells targeting EGFR loaded with miR-16-based mimic miRNAs, showed possible efficacy in MPM patients (5% had partial responses, and 68% exhibited stable disease). In fact, TargomiRs prevent uncontrolled tumor growth associated with the loss of the miR-15 and miR-16 family miRNAs [51].

The monoclonal ABT-806 antibody, a novel anti-EGFR antibody that selectively targets a unique epitope of the EGFR only exposed to overexpressed, mutant- or ligand-activated forms of the EGFR [52], could represent another attractive therapeutic strategy. Antibody–drug conjugates (ADCs) generated by conjugating ABT-806 to a cytotoxic payload (e.g., ABT-414, Depatux-M) proved effective in EGFR-amplified recurrent glioblastoma [53].

Preclinical data suggested that ABT-806-derived ADCs (ABT-414 and ABBV-322) also inhibit growth in MPM cell lines and are valid treatment options [54].

New strategies combining anti-EGFR target therapy and molecules against the other compensatory signaling pathways might become a future opportunity for this patient subgroup treatment.

However, no specific treatment approaches are available for patients with MPM and EGFR overexpression.

#### 3.2.2. AXL

The TAM family proteins member anexelekto (AXL) is a transmembrane tyrosine kinase receptor that is usually expressed in multiple solid tumors, including MPM [55]. It plays a crucial role in tumor development and metastatic spread [56], and it is considered a strong negative predictor of survival [57].

High levels of AXL expression lead to the activation of multiple signaling pathways and are related to drug resistance. In particular, AXL contributes to the suppression of the antitumor immune response and to the modification of the microenvironment, facilitating immune escape [58].

Recently, various clinical trials investigated the therapeutic potential of AXL’s inhibition [55].

Bemcentinib (BGB324) is an oral selective AXL inhibitor that in vitro suppresses cancer cell migration and invasion [59]. Based on preclinical studies suggesting that dual PD-1 and AXL inhibition is synergistic, the efficacy of Bemcentinib in association with pembrolizumab was evaluated in arm three of the phase IIa MiST umbrella trial, with encouraging results. Twenty-six patients were enrolled. The DCR at 12 weeks was 46.2% (90%CI 29.2–63.4), and ORR was 15.4% (95%CI, 4.4–34.9) with stable disease in 57.7% of cases; the DCR at 24 weeks was 38.5% (95%CI, 20.2–59.4) [60]. More recently, a novel therapeutic strategy combining the AXL inhibitors and inhibitors of ATR (kinases involved in cell cycle regulation and DNA replication and repair [61]) showed a promising synergistic effect on cell growth, apoptosis, and migration in MPM cell lines [62].

Due to the AXL and MET co-expression documented in some in vitro results, TKI multitarget inhibitors (e.g., cabozantinib) could play an important role in the treatment of MPM [63].

Therefore, regulating the AXL pathway may potentially improve the outcome of patients with MPM but more robust in vivo data are urgently needed.

### 3.3. Gene Involved in Hippo Signaling Pathway

#### 3.3.1. NF2, YAP1/TEAD

The neurofibromatosis type 2 (NF2) tumor suppressor gene was identified as a responsible gene for a familial cancer syndrome, neurofibromatosis type II, an inherited family cancer syndrome characterized by developing bilateral vestibular schwannomas [64]. NF2, located in chromosome 22q12, is frequently somatically mutated in MM. Non-sense/missense mutations or deletions with a loss of heterozygosity, and the consequent bi-allelic loss of function, in addition to gene rearrangements can be observed in up to 53% of MM; NF2 alterations are found more frequently in non-epithelioid MM [65,66]. NF2 mutation seems to be a late event, and it is linked to asbestos-induced genomic damage of NF2, resulting in more aggressive phenotypes [67].

The NF2 gene encodes Merlin (also called neurofibromin 2 or schwannomin), a moesin-ezrin-radixin-like 70 kDa protein belonging to the cytoskeletal linker protein family of Band 4.1 [68]. Merlin consists of three distinct domains: an N-terminal FERM domain (NTD), a central alpha-helical domain (CH), and a C-terminal tail domain (CTD). Oppositely to the other ezrin, radixin, and moesin (ERM) families of proteins, the actin-binding site in the C-terminal domain is missing in Merlin while there is a unique actin-binding motif in the N-terminal domain [69]. Merlin primarily localizes to the plasma membrane, mediating the contact-dependent inhibition of proliferation in normal cells. Conformational changes via phosphorylation or de-phosphorylation in the Merlin molecule regulate its open or closed form and thus the tumor suppressive activity. Merlin’s active form is thought to be regulated by dephosphorylation and lipid binding; interactions between Merlin, the plasma membrane, and the cortical actin skeleton determine tumor suppressor activity and regulate different cellular signaling pathways [70]. Multiple Merlin residues can be phosphorylated: the phosphorylation of Ser18 at its C-terminal tail is the most relevant in protein activity. Merlin indirectly links to cell adhesion molecules, receptor tyrosine kinases involved in the reception of extracellular signals, and downstream molecules that regulate intracellular signal transduction cascades (pro-oncogenic or tumor suppressive pathways, e.g., phosphoinositide 3-kinase (PI3K)/Akt, Hippo, and mammalian target of rapamycin (mTOR) pathways), regulating cell survival and proliferation [71,72,73].

The Hippo pathway seems to be a crucial signaling pathway linked to multiple aspects of cancer, and it is regulated via Merlin in mesothelial cells [74]. The four core components in this pathway, mammalian STE20-like protein kinase (MST1/2), Salvador homolog 1 (SAV1), MOB kinase activator 1A/B (MOB1), and large tumor suppressor kinase 1/2 (LATS1/2), all have tumor suppressive activity. The major targets of LATS1/2 kinases are transcriptional YAP and TAZ coactivators [74]. MM development and progression seem to be strictly related to the Merlin–Hippo pathway’s dysregulation. Hippo kinase core inactivation causes the inactivation of the LAST1/2 kinases, with the consequent dephosphorylation of YAP1 and TAZ; therefore, YAP and TAZ become activated and are translocated into the nucleus. Here, the activated forms regulate the transcription of numerous target genes that bind to different transcription factors, such as TEAD1-4 among others [75,76].

Therapeutic strategies in NF2-altered MM include mTOR/PI3K [77]. GDC-0980 (apitolisib), a PI3K and mTOR dual inhibitor, induced partial responses in MM patients in a phase I trial [78]. In the future, the Merlin–Hippo pathway’s influence must be taken into account for the development of mTOR inhibitors. Since YAP1 and TAZ are druggable targets, molecules that target YAP1/TAZ coactivators including TEADs’ interaction have been developed [79,80]. Verteporfin (Visudyne), a photosensitizer approved for macular degeneration treatment, was the first molecule developed and was shown to inhibit YAP1/TEAD interactions, diminishing YAP1 signaling [81]. Ongoing clinical trials include the first-in-class YAP/TEAD inhibitor VT3989; this molecule targets the Hippo pathway, inhibiting TEAD palmitoylation, which, in turn, blocks YAP function. The phase I dose escalation trial VT3989 showed promising results, with good tolerance and durable antitumor responses, in 69 patients, 46 of whom had malignant MM.

All these data, although arising from phase I trials, support the targeting of the Hippo–YAP–TEAD pathway [82].

#### 3.3.2. PI3K

The inhibitors of subunit PI3K-d are able to block PI3K/AKT activation with antitumor effects in breast cancer [83] and Merkel cell carcinoma [84]. The PI3K pathway and downstream proteins, which directly promote tumor cell survival and proliferation, are frequently activated in MM [85].

In the literature, there are few in vitro, proof-of-concept treatments with respect to MM cells with PI3K inhibitors. The vitro model roginolisib—an inhibitor of PI3K-δ—exhibited antitumor activity with respect to MM cells through PI3K-δ inactivation [86]. The authors detected the constitutive activation of the PI3K/AKT/mTOR signaling pathway in 74% of archival samples of MM, and they described antitumor and cell killing via roginolisib. Upregulated PI3K-δ expression in tumor cells appears to increase PI3K/AKT signaling. Moreover, PI3K-δ is preferentially expressed in immunosuppressive T regulatory cells, and its inhibition consequently enhanced effector T cell activity against tumor cells [84,87].

Other investigators reported a novel strategy for treating MM cell lines and primary culture cells from the pleural effusion of patients with MM. They treated the cells with CD4/6 inhibitors, obtaining a reduction in CDK6 and RB and the increased phosphorylation of AKT; then, the PI3K inhibitor blocked cell proliferation [88].

According to preclinical evidence, a phase I study with an inhibitor of class I PI3K isoforms, mTORC1/2 and DNA/PK and the Ly3023414 compound in patients with MPM and peritoneal MM and epithelioid, sarcomatoid, and mixed cells was proposed. The trial showed the limited activity of the drug, with three unconfirmed and one confirmed partial responses out of twenty-four patients, which did not favor further clinical research [89].

From the clinicaltrial.gov site, there is a phase I trial with a PI3K/mTOR kinase inhibitor VS-5584 administered in combination with FAK inhibitor VS-6063; however, the trial has already ended with no available reports provided. There are also two active phase I trials with PI3K inhibitors and IPI-549 (eganelisib) and AG01 compounds; however, there have been no updates thus far.

In conclusion, the inhibition of PI3K in MM, albeit with a strong underlying biological rationale, has currently not been given enough signals of clinical activity.

### 3.4. Enzyme Involved in Metabolism

#### 3.4.1. ASS1

Argininosuccinate synthetase1 (ASS1) is a urea cycle enzyme that catalyzes the condensation of citrulline with aspartate to form arginosuccinate, a precursor for a variety of molecules with important roles in tumorigenesis. ASS1 is frequently downregulated in MPM, and ASS1 loss is detected in 48–63% of cases, especially in biphasic or sarcomatoid histology [90].

In different solid tumors, including MM, lower ASS1 expression has been related to worse prognoses [90]. The lack of ASS1 expression could determine susceptibility relative to arginine deprivation due to the cells’ inability to synthesize arginine de novo (cells depend on exogenous arginine) [91]. Therefore, arginine deprivation was evaluated in several studies as a potential therapeutic target, focusing on the enzyme arginine deiminase (ADI), which is involved in arginine degradation.

In the randomized phase II ADAM trial, 68 patients with ASS1-deficient MPM received a pegylated modified form of this enzyme (ADI-PEG 20) versus BSC; PFS was 3.2 months vs. 2.0 months (HR 0.56, *p* = 0.03), and disease stability was observed in 52% vs. 22% of patients (no complete or partial responses were assessed) [91]. Notably, the greatest benefit was evidenced in patients with tumors that have a high degree of ASS1 loss (≥75%) and who were CT-naïve [92].

In addition, the clinical activity of combining ADI-PEG20 + cisplatin + pemetrexed was observed in the phase I TRAP trial: four (80%) of five patients with MPM included achieved a partial response (ORR 0.78; 95%CI, 0.39 to 0.97); the median OS was 56.4 weeks, and the median PFS was 30.7 weeks [93].

Similarly, in a larger dose-expansion cohort (N = 32) that only enrolled only patients with MPM, the DCR was 93.5% (95%CI: 78.6–99.2%), with a partial response rate of 35.5% (95%CI: 19.2–54.6%); the median PFS and OS were 5.6 (95%CI: 4.0–6.0) and 10.1 (95%CI: 6.1–11.1) months, respectively [94].

Recently, the randomized phase II/III ATOMIC-meso trial (NCT02709512) confirmed the efficacy of platinum + pemetrexed + ADI-PEG20 in 249 patients with non-epithelioid MPM compared with the placebo, ADI-PEG20 showed a higher mPFS [6.2 months (95%CI, 5.8–7.4) vs. 5.6 months (95%CI, 4.14–5.91); HR, 0.65; 95%CI, 0.46–0.90; *p* = 0.019] and a superior mOS [9.3 months (95%CI, 7.9–11.8) vs. 7.7 months (95%CI, 6.1–9.5); HR, 0.71; 95%CI, 0.55–0.93; *p* = 0.023]. A similar ORR was evidenced (13.8% vs. 13.5%, *p* = 0.95) [95].

Interestingly, anti-ADI-PEG20 antibodies were detected in 97.4% of patients by week 25 on pegargiminase, representing a possible potential resistance mechanism [95]. Preclinical descriptions identified an adaptive re-expression of ASS1 as another potential resistance mechanism relative to ADI-PEG20 [94].

Although these promising results suggested that personalized therapy targeting ASS1 may be possible, the survival benefit and the prognostic impact of ADI-PEG20 remain controversial.

Moreover, given the poor performance of CT instead of IO in the non-epithelioid histology MPM, data are still lacking regarding the comparison between this target approach and IO. The administration of ADI-PEG20 may be better suited for the post-IO setting.

#### 3.4.2. Glutamine

Glutamine metabolism is influenced by YAP1/TEAD signaling. The Krebs cycle, redox homeostasis, and the synthesis of nucleic acids all use glutamine as a substrate. SLC1A5 is the transporter of glutamine into cells [96]. The transcription activity of genes encoded for glutamine-metabolizing enzymes increases with the upregulation of glutamine metabolism via YAP1/TEAD signaling. Additionally, suppressing YAP1/TEAD signaling decreases SLC1A5 levels [97].

Preclinical results suggest that mesothelioma is dependent on glutamine and that glutamine depletion decreases YAP1/TEAD signaling. YAP1 levels and YAP1/TEAD target proteins can be decreased by limiting glutamine. V-9302 (an inhibitor of SLC1A5-dependent glutamine uptake) or CB-839, which inhibits the GLS-catalyzed conversion of glutamine to glutamate, has been developed for this purpose and demonstrated activity in MM cell lines [98]. Thus, limiting glutamine/glutamate could be an effective and viable treatment option for mesothelioma.

### 3.5. Surface Target

#### Mesothelin

Mesothelin (MSLN) is a cancer-associated antigen that is overexpressed on the membrane of cancer cells in several solid tumors including MM, especially in the epithelioid subtype [95,99,100,101,102,103,104]. The surface of healthy mesothelial cells of the pleura, pericardium, and peritoneum normally express MSLN in limited amounts [105]. The physiological function of MSLN expression in healthy tissues is little known [106] and it is supposed to be implicated in tumorigenesis, metastasis, and chemoresistance [99,107,108]. MSLN is initially expressed at the cell surface as a precursor protein of 71 kDa; the endoprotease Furin subsequently cleaves it, causing the release of megakaryocyte potentiating factor (MPF), which is a 31 kDa protein, and leaving MSLN in its mature form. Surface MSLN can also be released from the cell membrane by proteases, resulting in a soluble mesothelin-related peptide (SMRP) [109]. Three contiguous regions can be distinguished in I extracellular domain of MSLN: regions I (N-terminal region, residues 296–390), II (residues 391–486), and III (C-terminal region; residues 487–598) [110]. Region I, the membrane-distal region (MDR), can bind to the mucin MUC16 (alias CA125), which is highly expressed by the majority of MM cells; the MSLN–MUC16 interaction is important for adhesion and the promotion of cancer [111,112,113]. The blood and pleural fluid of MM patients revealed detectable MPF and SMRP. Over the years, SMRP and MPF have been evaluated as screening, diagnostic, prognostic, and predictive biomarkers for MM. SMRP but not MPF has an assay, the MESOMARK assay (a two-step immunoenzymatic assay in an ELISA format), that is the only FDA-approved blood test for MM [114]. The prognostic value of MSLN remains controversial [115,116,117,118,119]. SMRP assessments in the serum can be helpful for the tumor response assessment or predicting tumor progression as it reflects the tumor volume: the higher the tumor volume, the higher the SMRP levels [120,121]. Moreover, after surgery, serum SMRP levels decrease, and its longitudinal concentration measurements correlate with the tumor response [109,122].

As MSLN is highly expressed in cancer tissue and is low-to-non-existent in normal tissues, targeting it could reduce on-target/off-tumor toxicities; in addition, the high-level expression in the epithelioid MM and its association with tumor progression render MSLN an ideal biomarker and therapeutic target [123,124].

Various therapeutic approaches targeting MSLN have been assessed and are currently being tested in clinical trials. The MSLN MDR represents the main target for therapeutic strategies, due to the role of the MSLN–MUC16 interaction in tumorigenesis [125,126]. However, novel strategies are also targeting other MSLN regions [127,128].

MSLN-targeted therapies include monoclonal antibodies, ADCs, radio-immunoconjugates, T cell engagers, immunotoxins, and adoptive cellular therapies.

MORab-009 (amatuximab) is a monoclonal antibody of the chimeric IgG1 kappa type that targets the MDR region, inhibiting MSLN–MUC16 adhesion and stimulating cell lysis. In a phase I trial on advanced mesothelin-expressing (MSLN+) cancers, including MM, amatuximab in monotherapy demonstrated a good safety profile [129]. On this basis, amatuximab was investigated in combination with pemetrexed/cisplatin in a single-arm phase II study as first-line treatment on 89 patients with unresectable MM. An improvement in the OS rate (14.8 months) and disease control rate (DCR 90%, with 39.8% partial response and 50.6% stable disease; n = 83) compared with standard historical CT was reported. However, due to a 6-month PFS that was lower than the pre-set target (51.3% vs. 62%, respectively), the study did not meet its primary endpoint [130]. Subsequently, the randomized, placebo-controlled phase II ARTEMIS trial (NCT02357147) testing the combination of cisplatin and pemetrexed plus amatuximab/placebo was prematurely closed because of a business decision.

Anetumab ravtansine (AR) is a human anti-MSLN antibody (MF-T) conjugated to DM4, a tubulin inhibitory drug, and ravtansine [131]. Second-line AR or vinorelbine was evaluated in a randomized phase II trial. A total of 248 patients with MSLN+ MM (96% epithelioid subtype) progression during previous therapy were randomized at 2:1 relative to second-line AR or vinorelbine. However, a statistically significant difference between the experimental and standard therapy arms was not demonstrated both in PFS (4.3 months vs. 4.5 months, HR 1.22, *p* = 0.86) and OS (9.5 months vs. 11.6 months, HR 1.07, *p* = 0.66) [132]. To improve clinical efficacy, combinations of AR with other therapeutic strategies have been explored. In a phase Ib trial, the efficacy and safety of the combination of AR with standard first-line CT pemetrexed/cisplatin was investigated in MSLN+ MM (sixteen patients) and non-small cell lung cancer patients (one patient). The trial demonstrated an ORR of 50% (all PR) at the maximum tolerated dose (MTD) and a manageable safety profile [133]. The safety and efficacy of AR in combination with pembrolizumab in MPM patients will be assessed in a phase I/II trial (NCT03126630); the trial is active but not recruiting. Another phase II trial (NCT03926143) is terminated.

BMS-986148 is an anti-MSLN antibody conjugated with tubulysin, a cytotoxic drug. Some efficacy of this treatment in MM patients was observed in preliminary data—ORR of 4% for monotherapy and 31% ORR for the combination, with durable responses (up to 9 months) [134].

MSLN-TTC or BAY2287411 is a radio-immunoconjugate consisting of a fully human anti-MSLN antibody linked to the alpha-emitting radioisotope thorium-227 via a covalently chelating agent [135]. A phase I clinical trial (NCT03507452) investigating the safety and activity of BAY2287411 in MM and ovarian cancer patients, who have exhausted available treatment options, was completed.

Immunotoxins include an antibody linked to a bacterial toxin that, once internalized by tumor cells, determines the inhibition of protein synthesis. The SS1P immunotoxin comprised a murine anti-mesothelin antibody that binds to the MSLN MDR and is conjugated to a Pseudomonas exotoxin (PE) fragment. In a phase I trial, the combination of SS1P with platinum-based standard CT was investigated; despite CT-induced myelosuppression, neutralizing anti-drug antibodies (ADAs) were detected in almost all patients [136]. The combination of SSP1 with pentostatin and cyclophosphamide has demonstrated clinical efficacy, and ADA formation was markedly delayed [137].

LMB-100 is a humanized anti-MSLN Fab fragment (avoiding the formation of ADAs) linked to a PE toxin that is less immunogenic: PE24. A phase I trial evaluated LMB-100 in patients with solid MSLN+ tumors, including ten MM cases that progressed relative to platinum CT, which did not demonstrate tumor responses. Despite expectations, the development of ADAs was observed in all patients after repeated administrations [138]. Ten patients were treated with IO after progression to LMB-100, and four of them exhibited durable responses [139]. Based on these promising results, two cycles of LMB-100 followed by pembrolizumab for up to 2 years were administered to MM patients who progressed on platinum CT in a phase II trial (NCT03644550). Clinical outcomes from this trial will help determine if the combination of anti-MSLN immunotoxins and IO could be a possible treatment for MSLN+ tumors.

Due to the growing interest in cancer vaccines caused by the encouraging results seen in other malignancies, this approach was evaluated also in MM. Cancer vaccines can elicit T cells inducing a specific and powerful antitumor immune response with less adverse reaction in normal tissues than the other immunological treatments [140].

MSLN-directed vaccination consists of the use of Listeria monocytogene-expressing MSLN vaccine (LM-mesothelin), CRS-207. The aim of this cancer vaccine is to boost immunity against MSLN-expressing tumor cells [141]. After demonstrating a good safety profile in a phase I trial [142], CRS-207 was tested as a monotherapy or combined with cyclophosphamide, with promising objective tumor responses [143]. Another novel chimeric DNA vaccine generated using antigen-specific connective tissue growth factor lined and MSLN (CTGF/MSLN), then combined to immuno-modulators, showed a potent antitumor effect in MM [144].

A recent potential strategy includes adoptive cell therapy targeting MSLN [145]. Chimeric antigen receptors (CARs) are engineered proteins expressed on the surface of T cells aimed to target tumor cells. The typical structure consists of an ectodomain, containing a single-chain variable fragment (scFv) that binds to a specific tumor antigen (in this case, MSLN), a hinge, a transmembrane domain, and an endodomain with the signaling domains. As the CAR T cell persists in the body and reactivates in the case of subsequent antigen encounters, this new treatment paradigm promotes immune surveillance and avoids tumor recurrence [146]. CAR T cell efficacy, without toxicity effects, was demonstrated in numerous preclinical studies [147,148]. A CAR T cell product using mRNA electroporation that transiently expressed the anti-MSLN CAR on T cells, exhibited a safety profile in a phase I trial; however, no tumor responses were registered in patients with MM [149]. The same CAR with a lentivirus vector was evaluated in a second phase I trial of the same group: 11 (out of 15) patients exhibited a stable disease 28 days post-infusion, but 5 of them progressed later [150]. The most recent phase I trial by the UPenn group is active but not recruiting. CAR T cells are administered intravenously and locoregionally directly into the pleural space. The goal of locoregional delivery is to increase efficacy, overcoming the barriers of tumor stroma. Multiple trials have been developed administering CAR T intrapleurally in pretreated MM (NCT03608618, NCT02414269, and NCT04577326) because locoregional delivery has resulted in more effective cancer control in preclinical studies [151]. Memorial Sloan Kettering conducted a single-center, open phase I/II study (NCT024142699) with locoregional delivery, demonstrating a partial response in 2 of the 16 patients and a stable disease in 9 patients, without major toxicities. A subset of patients also received pembrolizumab, suggesting a possible synergism in combining CAR T cells with IO [152,153]. T cell receptor fusion constructs (TRuCs) conjugate the antigen-specific scFv to the N-terminus of the CD3e T cell receptor complex. A phase I clinical trial (NCT03907852) is testing this new strategy. Very encouraging preliminary results were presented at AACR 2021: tumor regression in all eight treated patients was achieved [154]. Two trials are investigating CAR T cells carrying a PD-1-dominant negative receptor (NCT04577326) or one that is modified to secrete anti-PD-1 nanobodies (NCT04489862).

### 3.6. Genes Involved in Responses to DNA Damage

#### 3.6.1. BAP1 and EZH2

Genomic alterations in MPM are primarily related to the loss of function of tumor suppressor genes. The breast cancer gene 1 (BRCA1)-associated protein 1 (BAP1) is the most commonly altered gene (approximately 60% of MPM; mutated, deleted, or epigenetically silenced) [36]. It is located on chromosome 3p21 and encodes the catalytic core of the polycomb repressive deubiquitinating complex. BAP1 inactivation is almost always somatic, but 3–6% MPM arise from germline BAP1 mutation.

BAP1 inactivation increases the expression of enzyme enhancer zeste homolog 2 (EZH2), a component of the histone methyltransferase polycomb repressive complex 2 (PRC2). This complex leads to chromatin remodeling catalyzing the trimethylation of histone H3 on lysine 27 (H3K27me3) [155]. EZH2 is an oncogenic driver that regulates gene expression, and it is required for the physiological differentiation of lung mesothelium. It plays a key role in silencing epigenetic genes; indeed, its dysregulation is associated with carcinogenesis [156,157]. Furthermore, several studies have highlighted that EZH2 may promote the activation of key oncogenic programs through its direct interaction with transcription factors [158]. The overexpression of EZH2 in MPM specimens has been related to aggressiveness and poor prognosis [159]. Recently WHO recognized EZH2 as a diagnostic marker allowing to distinguish MPM from benign mesothelial proliferation [160].

In preclinical models, the BAP1 mutant MM cell lines exhibited enhanced sensitivity to EZH2 inhibitors. Even in the xenograft mouse model, increased activity was observed in BAP1 mutant tumors compared with wild-type MM [155].

To our knowledge, the only currently published clinical trial on the use of an EZH2 inhibitor on MPM is that of Zauderer and colleagues [161]. It is a multicenter, international, open-label, single-arm phase II study that evaluated the use of tazemetostat in relapsed or refractory patients with BAP1-mutated MPM to at least one pemetrexed-containing regimen (74 patients). Molecular evidence of the BAP1 loss of function was assessed via the immunohistochemical determination of the absence of BAP expression in the nucleus. In part 2 of the trial, 61 patients were treated with tazemetostat 800 mg (200 mg tablets) twice daily. The disease control rate at week 12 (primary endpoint of part 2) was 54%, decreasing to 33% at week 24. Only two patients (3%) had a partial response. In the overall population, the mPFS was 18 weeks and mOS was 36 weeks. The most common treatment-related adverse events were fatigue, decreased appetite, dyspnea, and nausea. One of the major biases is that a substantial proportion of patients underwent previous surgical resection, which is not reflective of real-life patients.

One phase I/II clinical trial is recruiting patients with advanced solid tumors (including MM) and lymphomas, and they receive monotherapy treatment with CPI-0209, an EZH2 inhibitor (NCT04104776).

In view of the modest activity of EZH2 inhibitors as single agents, therapeutic combinations have been tested in preclinical studies.

Recently, Badhai and colleagues demonstrated a high synergy of action of the combination of FGFR and EZH2 inhibitors on mutated BAP1 MM cell lines [162]. This synergy has been confirmed in in vivo studies on mice. The combination of ATM and EZH2 inhibitors also appears to be synergistic in BAP1-deficient MM [163].

EZH2, being an epigenetic modulator, also has an effect on the tumor microenvironment. The inhibition of EZH2 could increase the immunogenicity of tumor cells by redefining cellular epigenetic structure and favoring the expression of genes, coding for both the presentation of new antigens and the recruitment of antitumor immune cells [164,165]. In an MPM multicellular spheroid model (MCS), the use of tazemetostat led to an increase in the expression of chemokines for cytotoxic immune cells (e.g., CXCL9 and CXCL10) and monocyte (e.g., CCL2, M-CSF, CCL5, CXCL12), modifying the TME composition. Based on preclinical studies, EZH2 inhibitors may act synergistically with IO. This association could be a new potential therapeutic approach.

To our knowledge, there are currently no active combined treatment clinical trials on MPM.

#### 3.6.2. BRCA

BAP1 binds to BARD1 to form a BRCA1–BARD1–BAP1 complex that is involved in the homologous recombination (HR)-mediated repair of double-strand DNA breaks (BSBs). PARP enzymes are essential for the repair of single-strand DNA breaks. PARP inhibition results in single-strand break accumulation that could become BSBs, which are lethal in HR repair-deficient cells [166,167]. BRCA1 (breast cancer susceptibility gene 1) and BRCA2 are tumor suppressor genes, and mutant phenotypes are predisposed to breast and ovarian cancers [168]. Even if the role of BRCA1 in MM remains to be elucidated, as both BAP1 and BRCA1 are involved in the DNA damage response, they can be considered biomarkers for targeted therapies with PARP inhibitors (PARPi). The antiproliferative effect of PARPi in MM cell lines with BAP1 alterations was observed in vitro studies [169]. Based on mixed preclinical results [170,171], two clinical trials have evaluated PARPi monotherapy in patients with MM.

The MiST1 study was a single-arm phase IIa trial that enrolled MM patients with BAP1-deficient or BRCA1-deficient MM pretreated with chemotherapy [172]. In total, 26 subjects received PARPi rucaparib. The 12-week DCR was 58%, mPFS was 17.9 weeks, and mOS was 41.4 weeks. Only three partial responses were found.

Ghafoor and colleagues conducted another phase II single-arm, open-label study in pretreated MM patients using the PARPi olaparib [173]. A total of 23 patients, irrespective of BAP1 or BRCA1 status, were enrolled (including 7 with peritoneal MM). There was no objective response, and the mPFS and mOS were 3.6 months and 8.7 months, respectively. Efficacy was independent of BAP1 mutation, even with the germline BAP1 mutation as a negative predictive factor of PARP inhibition response [174].

Thus, the antitumor activity of PARPi monotherapy in MM patients seems to be limited. Furthermore, the immunohistochemistry pattern of BRCA1 or BAP1 was not directly associated with the response to PARP inhibitors, indicating that other mechanisms likely contribute to PARP inhibitor sensitivity [170,172,173].

It has been established that BAP1 is linked to an inflamed tumor microenvironment and the infiltration of cytotoxic T cells, suggesting a possible synergistic activity of the combination of IO and PARPi [174,175].

Unfortunately, the results of the recent interim analysis of the prospective UNITO-001 phase II study are not encouraging [176]. The study aimed to investigate the combination of the PARPi niraparib plus the anti-PD-1 dostarlimab in pretreated patients with homologous recombination repair deficiencies (defined as the presence of somatic or germline mutations in the DNA homologous recombination repair pathway), PD-L1 > 1% non-small cell lung cancer, and MPM. Only 17 of the 183 screened patients were included (12 MPM and 5 non-small cell lung cancer). mPFS, the primary object, and mOS were 3.1 and 4.2 months, respectively, with an ORR of 6% (1/17 patients, notably one with BRCA2 mutation). To note, mPFS was lower (2.9 months) in the cohort of BAP1 mutant MPM. Contrarily, the patient with the BAP1 germline mutation exhibited signs of sustained activity in terms of stable disease.

The data suggest that the use of PARPi, although backed by a strong biological rationale, still requires better patient selection and an understanding of potential biomarkers.

## 4. Discussion

Advanced MM is a highly aggressive and lethal disease, with little therapeutic progress in recent decades. CT has been the backbone of first-line treatment for MM for the last three decades. While the introduction of IO resulted in an increase in survival compared to standard CT and ipilimumab–nivolumab has emerged as the new first-line treatment, IO is not approved for epithelioid histology in many countries. Nonetheless, the prognosis remains poor, with a median survival of only 18 months. Furthermore, MM continues to be an orphan pathology concerning post-first-line therapy as no validated treatment is available. Therefore, there is a desperate clinical need to develop new therapeutic strategies while also refining the targeted approach.

As discussed above, the rapid improvement in understanding MM biology and the genome-wide characterization of pathways altered in MM patients led to the development of novel therapeutic targets in in preclinical trials in order to transfer them to clinical settings. Various phase I/II studies are testing new therapies for the most promising targets, including CAR T and cancer vaccines versus MSLN, PRMT5-MTA, PI3K, andYAP/TEAD inhibitors and PARPi (Table 1).

Beyond the described target and relative therapeutic attempts described above, rare agnostically druggable alterations of ALK, NTRK, KRAS, ERBB2, and FGFR have also been described in MM [31]. In particular, ALK rearrangement was described in 0.36% and 1.13% of MPM and peritoneal MM, respectively [31], with a higher incidence in patients younger than 40 years, irrespective of the site of the disease [177]. In another cohort, NTRK and ALK rearrangements in MPM were reported in 0.6% of cases [178]. Growing evidence is accumulating, albeit only through case reports, about the potential effectiveness of ALK tyrosine kinase inhibitors (TKIs) in this MM subgroup [179,180]. In addition, KRAS G12C was described in approximately 1% of MPM [1,8], suggesting a potential agnostic role of KRAS TKI. The rarity of such activating mutations once again stresses how the molecular landscape of MPM is characterized by inactivating tumor suppressor alterations [29,31]. Regarding surface targets, the oncofoetal glycoprotein 5T4 could represent another valid antigen for targeted therapies due to its wide expression on mesothelioma cell lines in all MM subtypes. In the phase II SKOPOS trial combination of pemetrexed–cisplatin chemotherapy and TroVax, a viral vaccination containing the 5T4 glycoprotein gene revealed robust immune activity and efficacy (mOS 10.9 months) with acceptable safety and tolerability [144].

Although initial clinical data are encouraging, the treatment’s stratification via molecular characteristics for MM is only at its beginning. Despite advances in understanding the molecular biology of MM, to date, there have been relatively few changes in standard clinical practice based on these findings. Actually, histology remains the primary tool in determining treatment stratification, but this did not translate into significant survival gains. Given the low response rates of the mentioned therapies in monotherapy and the complexity of the biology of MM, combination treatments—for example, with IO or with existing approved CT regimens for MPM—are also another possible future alternative. In addition, new therapeutic promising approaches like CAR T and cancer vaccines targeting a series of self-antigens commonly overexpressed in MM could enlarge the therapeutic landscape.

Undoubtedly, complete definitions of phenotypes of MM and pathogenetic mechanisms underlying its evolution are still largely nebulous and unknown, and there is definitely a heterogeneity that the histological classification fails to grasp. Early attempts of multiomic analyses—namely, the integration of multiple biomarkers of MPM—suggest that a classification of MPM based on simultaneous morphology assessments, genomics, and factors related to methylation and the immune system may capture a more precise frame of the disease [181].

## 5. Conclusions

More in-depth knowledge of potential targetable mutations in MM is fundamental in order to widen the therapeutic option panorama in this rare and desperately lethal disease. In recent years, many new strategies have emerged as a hope for patients with MM, which inexorably continue to have disheartening outcomes. A broad adoption of molecular analysis in MM should be implemented and clinical trials encouraged, as the road to precision medicine in MM, although promising, still faces uphill battles.

## Figures and Tables

**Figure 1 cancers-16-01971-f001:**
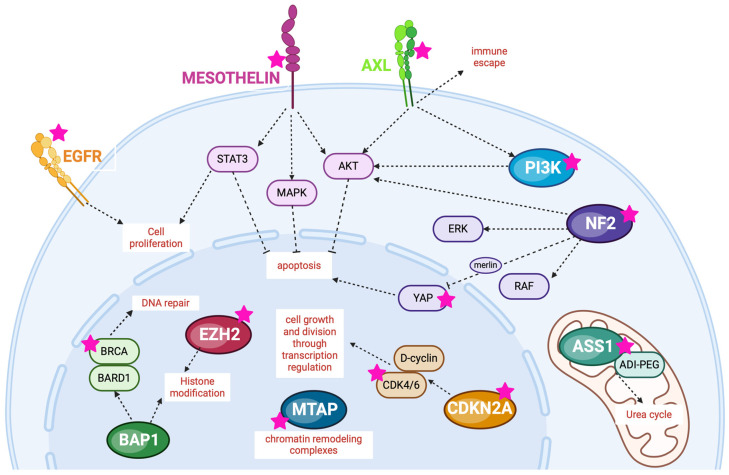
Main targets and pathways. Created with BioRender.com (accessed on 19 May 2024).

**Table 1 cancers-16-01971-t001:** Main ongoing trials with targeted therapies alone or combined with chemo/immunotherapy in MM.

NCT Identifier	Phase	Drud(S) Class	Population	Treatment Arms	Status	Primary Endpoint
NCT03054298	I	Anti-MSLN CAR T cells	MSLN-expressing tumors	HuCART-meso	Active, not Recruiting	Number of participants with treatment-related AEs
NCT02414269	I/II	Anti-MSLN CAR T cells	MPM Lung cancer Breast cancer	Genetic: iCasp9M28z T cell infusions Drug: pembrolizumab	Active, not Recruiting	Phase I: composite measure of severity and number of AE changes Phase II: clinical benefit rate
NCT03907852	I/II	Anti-MSLN CAR T cells	MM, cholangiocarcinoma, ovarian cancer, NSCLC	Biological: gavo-cel Drug: fludarabine/cyclophosphamide /nivolumab/ipilimumab	Active, not Recruiting	Phase I: DLTs within 28 days post-treatment Phase II: ORR at 3 months; DCR based on ORR + SD lasting at least 8 weeks
NCT06256055	I	Anti-MSLN CAR T cells	Colorectal cancer MM Bile duct cancer Rectal cancer Ovarian cancer Pancreatic cancer Breast cancer	UCMYM802 injection	Active, recruiting	Treatment-emergent adverse event (TEAE) Treatment-related adverse event (TRAE) Adverse events of special interest (AESI) Incidence of dose-limiting toxicities (DLTs)
NCT05795595	I/II	Anti-MSLN CAR T cells	Clear cell renal cell carcinoma Cervical carcinoma Esophageal sarcinoma Pancreatic carcinoma MPM	CTX131	Active, recruiting	Phase I: incidence of AEs Phase II: ORR
NCT06051695	I/II	Anti-MSLN CAR T cells	Solid tumor Colorectal cancer NSCLC Pancreatic cancer Colorectal adenocarcinoma Ovarian cancer MM	A2B694	Active, recruiting	Phase I: rate of adverse events and DLTs by dose level + recommended phase II dose Phase II: ORR
NCT04577326	I	Anti-MSLN CAR T cells	MPM	Drug: cyclophosphamide Biological: CAR T cells	Active, recruiting	MTD
NCT02637531	I	PI3K inhibitor	Advanced solid tumors, for part G: relapsed MM, any histology	IPI-549	Active, recruiting	Antitumor activity
NCT05627960	I	PI3K inhibitor	Triple-negative breast cancer Hormone-resistant breast cancer Non-small cell lung cancer, MM	AG01	Active, recruiting	DLT and antitumor activity
NCT05245500	I/II	PRMT5-MTA inhibitors	Advanced solid tumors with homozygous MTAP deletion	MRTX1719	Active, recruiting	Phase I: number of patients who experience dose-limiting toxicity and TRAE Phase II: ORR, DOR, PFS, and OS
NCT05275478	I/II	PRMT5-MTA inhibitors	Advanced solid tumors with homozygous MTAP deletion	TNG908	Active, recruiting	Phase I: MTD and dosing schedule of TNG908 Phase II: efficacy by RECIST or mRECIST v1.1 or modified RANO criteria
NCT05732831	I/II	PRMT5-MTA inhibitor	MTAP-deleted solid tumors	TNG462	Active, recruiting	Phase I: MTD and dosing schedule Phase II: antineoplastic activity
NCT05455424	II	PARPi	Relapsed or refractory MM	Niraparib vs. active symptom control	Active, not recruiting	PFS
NCT04940637	II	PARPi + anti-PD-1	Relapsed or refractory MM	Niraparib + dostarlimab	Active, recruiting	PFS
NCT04515836	II	PARPi	Relapsed MM with BAP1 loss or mutations in cells that disrupt protein function	Olaparib	Active, recruiting	ORR
NCT04665206	I	YAP/TEAD inhibitor	Relapsed MM or solid tumors NF2-mutated	VT3989	Active, recruiting	DLT and occurrence of general toxicity
NCT04857372	I	YAP/TEAD inhibitor	Relapsed MM or solid tumors NF2-mutated or with YAP/TAZ fusion	IAG933	Active, recruiting	DLT, number of SAE, and number of patients with dose interruption/changes
NCT05765084	I/II	PD-L1 inhibitor + WT1/DC vaccination	I line epithelioid MPM	Atezolizumab + WT1/DC vaccines + platinum/pemetrexed	Active, recruiting	Proportion of patients that experienced (S)AEs, number and grade of AEs and SAEs, and proportion of patients who completed study treatment schedule
NCT03126630	I/II	Anti-MSLN + anti-PD-1	MSLN + MPM	Anetumab ravtansine + pembrolizumab	Active, not recruiting	Phase I: safety dose of anetumab ravtansine Phase II: ORR combination vs. pembrolizumab
NCT04287829	II	Anti-VEGFR + anti-PD-1	II and III line MPM	Lenvatinib + pembrolizumab	Active, recruiting	ORR
NCT05425576	II	TGF-b2 inhibitor + anti-PD-1	MPM failing to respond to checkpoint inhibition	OT-101 + pembrolizumab	Active, not recruiting	ORR
NCT06031636	II	Oncolytic virus + anti-PD-1	Advanced MPM resistant to advanced PD-1 inhibitors	Oncolytic adenovirus H101 + pembrolizumab	Active, recruiting	ORR, DCR
NCT04013334	II	Cancer vaccine + anti-PD-1	Relapsed MM	MTG201 (intratumoral injection) + nivolumab	Active, not recruiting	ORR
NCT04040231	I	Cancer vaccine + anti-PD-1	WT1-expressing MPM	Galinpepimut-S + nivolumab	Active, not recruiting	MTD

Abbreviations: AEs = adverse events; DLT = dose-limiting toxicity; MTD = maximum tolerated dose; SAEs = serious adverse events; DOR = duration of response; and MM = mesothelioma.

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
