# Peer review of "Targeted Therapy in Mesotheliomas: Uphill All the Way"

_cancers, 2024, doi:10.3390/cancers16111971_

Round 1
Reviewer 1 Report
Comments and Suggestions for Authors
In the introduction please revise the nomenclature according the "2021 WHO Classification of Tumors of the Pleura". Mesothelioma, pleural mesothelioma, MM, MPM are used inappropriately throughout the paper. The paper must be completely revised for English. Moreover, there are many typos, lowercase letters, uppercase letters etc. etc...(e.g. line 114, 116, 119, 367, 417....). The text must be carefully revised. Trials in NSCLC or SCLC, cited in 8.1, are not relevant for this review, please remove. Change the title "genes related to receptors tyrosine kinase" in "genes coding for.." Modify also the title "enzymes related to metabolic cycles" in "enzyme involved in metabolism". In the chapter "gene involved in DNA repair" it is reported the trial with the EZH2 inhibitor, tazemetostat, but the role of EZH2 in mesothelioma is not restricted to DNA repair; authors report the sinergy with the ATM inhibitor, but they focus more on the role of EZH2 in TME.
Comments on the Quality of English Language
The paper must be completely revised for English.
Reviewer 2 Report
Comments and Suggestions for Authors
The authors in the present article have highlighted a comprehensive review on the current scenario of the treatment associated with the rare disease of mesothelioma cancer. The review article discusses successfully on a broad view to examine the main targets and the related evidence regarding possible treatment activities intended for them.
The article is very well written and portrays a highly descriptive and comprehensive study.
Few points that can be addressed to enrich the present study includes,
1. If the authors can include some facts on the causes of the mesothelioma cancer with the incidence rate.
2. Is age a factor related to mesothelioma cancer prevalence.
3. Do the authors have any idea on the combination therapies that are in the clinical trials along with the targeted therapies IO. If possible, to list them all in the table.
4. Can the authors include any information on the cancer vaccines approaches in mesothelioma.
Comments on the Quality of English Language
Minor editing
Reviewer 3 Report
Comments and Suggestions for Authors
The review focused target therapies for malignant mesothelioma (MM), including different potential targets of tumour cells of MM. The authors described the targets and related clinical trials. MMs are featured with mutations in tumour suppressor genes, so the target therapies are difficult to develop. It’s good to give an overview of attempts for target therapy development in MMs. There are major comments as follow:
1. The authors need to proofread the text to avoid spelling mistakes. For example, ‘the search…’ in line 11 should be ‘The search…’. In line 163, ‘selectiv’ should be ‘selective’. Line 364, ‘beimplicated’ should be ‘be implicated’. Line 367, ‘form..’ should be ‘form.’, where one dot was redundant. Line 417, ‘inMSLN+…’ should be ‘in MSLN+’. Line 309, ‘In conclusion the…’ should be ‘In conclusion, the…’, where a comma is missing. Besides, the text font size is not consistent in the Table 1, as the text in PI3K inhibitor rows seems smaller than others.
2. One suggestion for the title, is it better to change into ‘Target therapies in malignant mesotheliomas: uphill all the way’? Because precision medicine means personalized treatment for different patients, emphasizing biomarkers for different treatment. In this review, the authors mainly talked about target therapies for MM, not stratifying patients and choose different treatment options.
3. Regarding to the subtitles, I suggest merge 3-8 into ‘3. Potential molecular targets for MMs’. Then ‘3. Gene involved in cell-cycle regulation’ will be ‘3.1 Genes involved in cell-cycle regulation’. ‘4. Gene related to RTKs’ will be ‘3.2 Genes related to RTKs’, etc. ‘9. Discussion’, and ’10. Conclusions’ will be ‘4. Discussion’, and ‘5. Conclusions’, respectively.
4. For the paragraphs, the authors should adjust them, as it’s no need to keep one sentence as one paragraph. For example, sentences can be one paragraph from line 60 to line 70, as they are all related chemotherapy. Line 208 should be together with the following paragraph (Line 210-216), right? The references for line 208 are missing, clinical trials for AXL inhibitors in MMs. The paragraph (line 228-266) is too long. Suggest divide into 3 paragraphs, one paragraph talking NF2 (line 228-236), second is describing merlin (line 237-255), and the third is about pathway cascades regarding YAP (line 256-266).
5. In the discussion and table 1, NF2 inhibitor was included as promising target therapy. What’s the rationale? Because NF2 is a tumour suppressor gene, inhibition of NF2 will promote tumour growth, in theory.
6. Glutaminase inhibitor seems a promising target therapy drug for MMs. The authors should include it in the metabolic part. The paper ‘Mesothelioma cancer cells are glutamine addicted and glutamine restriction reduces YAP1 signaling to attenuate tumor formation’ can be referred.
Comments on the Quality of English Language
Fine. Understandable.
Round 2
Reviewer 1 Report
Comments and Suggestions for Authors
The authors consistently revised their paper that is now suitable for publication
Author Response
THANK YOU
Reviewer 3 Report
Comments and Suggestions for Authors
'3.4.2 Glutamine' should be '3.4.2 Glutaminase'. Besides, this part should be under 3.4.1 ASS1, not below 3.5.1
Author Response
'3.4.2 Glutamine' should be '3.4.2 Glutaminase'. Besides, this part should be under 3.4.1 ASS1, not below 3.5.1
THANK YOU FOR POINTED THIS OUT. WE HAVE DANE THE CORRECTION